# Effect of Biogas Slurry on the Soil Properties and Microbial Composition in an Annual Ryegrass-Silage Maize Rotation System over a Five-Year Period

**DOI:** 10.3390/microorganisms12040716

**Published:** 2024-04-01

**Authors:** Guangyan Feng, Feixiang Hao, Wei He, Qifan Ran, Gang Nie, Linkai Huang, Xia Wang, Suhong Yuan, Wenzhi Xu, Xinquan Zhang

**Affiliations:** 1College of Grassland Science and Technology, Sichuan Agricultural University, Chengdu 611130, China; feng0201@sicau.edu.cn (G.F.); haofx412828@163.com (F.H.); nieg17@sicau.edu.cn (G.N.); huanglinkai@sicau.edu.cn (L.H.); wangxia@sicau.edu.cn (X.W.); 15182854854@163.com (S.Y.); 2Industrial Crop Research Institute, Sichuan Academy of Agricultural Science, Chengdu 610066, China; 3Grassland Research Institute, Chongqing Academy of Animal Science, Chongqing 402460, China; cqhewei1978@163.com (W.H.); ranqifan@outlook.com (Q.R.)

**Keywords:** biogas slurry, bacterial community, fungal community, soil chemistry, annual ryegrass-silage maize, soil

## Abstract

Soil health is seriously threatened by the overuse of chemical fertilizers in agricultural management. Biogas slurry is often seen as an organic fertilizer resource that is rich in nutrients, and its use has the goal of lowering the amount of chemical fertilizers used while preserving crop yields and soil health. However, the application of continuous biogas slurry has not yet been studied for its long-term impact on soil nutrients and microbial communities in a rotation system of annual ryegrass-silage maize (*Zea mays*). This study aimed to investigate the impacts on the chemical properties and microbial community of farmland soils to which chemical fertilizer (NPK) (225 kg ha^−1^), biogas slurry (150 t ha^−1^), and a combination (49.5 t ha^−1^ biogas slurry + 150 kg ha^−1^ chemical fertilizer) were applied for five years. The results indicated that compared to the control group, the long-term application of biogas slurry significantly increased the SOC, TN, AP, and AK values by 45.93%, 39.52%, 174.73%, and 161.54%, respectively; it neutralized acidic soil and increased the soil pH. TN, SOC, pH, and AP are all important environmental factors that influence the structural composition of the soil’s bacterial and fungal communities. Chemical fertilizer application significantly increased the diversity of the bacterial community. Variation was observed in the composition of soil bacterial and fungal communities among the different treatments. The structure and diversity of soil microbes are affected by different methods of fertilization; the application of biogas slurry not only increases the contents of soil nutrients but also regulates the soil’s bacterial and fungal community structures. Therefore, biogas slurry can serve as a sustainable management measure and offers an alternative to the application of chemical fertilizers for sustainable intensification.

## 1. Introduction

Silage crop yields play a key role in the production of livestock. In the last few decades, chemical fertilizers have effectively enhanced the productivity of silage crops by improving the fertility of soil. Previous research has shown that the application of nitrogen (N) fertilizer can increase the production and crude protein content of forage grass, thereby improving its nutritional quality [1]. However, excessive fertilization with N has a negative impact on the growth and yield of crops [2,3]. It was previously reported that the application of N fertilizer may lead to a reduction of 3.3% to 14.2% in the crude protein concentration of forage dry matter. Interestingly, the negative impact on the levels of crude protein owing to the highest N fertilization application can be mitigated by the moderate application of phosphorus fertilizer [4]. In addition, excessive N application can lead to lower crop yields and soil acidification [5,6]. Moreover, the long-term application of inorganic N fertilizer reduces the relative abundance of the soil’s bacterial and fungal communities [7]. Therefore, selecting the appropriate fertilization method is an important management strategy to maintain soil health and achieve sustainable development [8].

Biogas slurry is mainly formed from the highly concentrated organic wastewater produced by the anaerobic fermentation of organic matter, such as livestock and poultry manure [9]. Currently, biogas slurry is extensively utilized as a potential organic fertilizer to enhance soil fertility and agricultural productivity, owing to its abundant nutrients. The beneficial effects of biogas slurry on plant growth and soil properties and on microorganisms have been demonstrated in most field trials [10,11,12]. The application of biogas slurry not only improves the physical and chemical characteristics of the soil but also effectively regulates the structure of soil bacterial communities and enhances the diversity of soil fungal communities [13,14,15]. However, previous research has shown that compared with the application of low-concentration biogas slurry (250 m^3^ ha^−1^ yr^−1^), the application of high-concentration biogas slurry (375 m^3^ ha^−1^ yr^−1^) yielded the highest alpha diversity in the fungal community, but not in terms of bacterial diversity. Furthermore, the application of biogas slurry also significantly influenced the abundance of functional fungi such as *Aspergillus*, *Trichoderma*, and *Penicillium* [16]. The application of biogas slurry over an extended period of time can increase the loss of N and the emissions of nitrous oxide (N_2_O) from the soil, owing to nitrification [17]. Previous research has shown that the yield in a maize-wheat (*Zea mays*-*Triticum aestivum*) rotation can be maximized by applying 226 kg N ha^−1^ and replacing it with 38% biogas slurry [18]. Moreover, the yield of rice (*Oryza sativa*) can be sustained with a replacement ratio of 50% biogas slurry and an application of 270 kg N ha^−1^ [19]. Alternatively, China is the fifth-largest livestock producer in the world and it is estimated that over 551 Mt of manure (dry weight base) is generated in China per year. Therefore, the effective utilization of biogas slurry needs further attention [20,21]. The application of biogas slurry, rather than chemical fertilizer, can solve the issue of the excessive manure produced by animal husbandry. Additionally, it is recognized as a key factor in achieving a sustainable biogas slurry-forage grass–livestock production cycle [22,23,24]. However, the improper application of biogas slurry may cause soil salinization and heavy metals accumulation, posing a potential pollution risk to farmland soil [25]. Therefore, it is necessary to study the effects of biogas slurry application on forage planting soil.

As an essential part of the soil ecosystem, soil microorganisms are involved in the decomposition of animal waste and plants [26], the formation of humus, and the transformation and cycling of nutrients [27,28]. Soil microorganisms play a crucial role in maintaining soil fertility and soil microbial communities, which are the indicators used for evaluating soil health [29]. Soil microorganisms facilitate numerous soil processes, such as the cycling of nutrients, the breakdown of organic matter, and the interactions between plants and microbes. Studies have shown that microbial communities are essential to control nutrient cycling; they also affect plant productivity and maintain the stability of ecosystems [30,31,32,33]. The application of biogas slurry can increase soil pH and NH_4_^+^-N, mitigate soil acidification, and reshape the soil’s microbial community [34]. To date, most studies have focused on the short-term effects of biogas slurry on yield and quality in forage monoculture systems [35,36,37]. Therefore, long-term field experiments are needed to thoroughly investigate the changes in soil properties and microbial communities in order to unravel the complexity of biogas slurry application to soil.

In this study, a field experiment was conducted over a period of 5 years that examined the rotation of annual ryegrass and silage corn in individual plots. The aim was to investigate the effects of the long-term application of biogas slurry, chemical fertilizer, and their combination on the soil’s chemical properties and soil microbial community composition. In particular, the following research questions were our focus: (1) After a period of 5 years, what changes can be observed in the diversity and composition of the soil bacterial and fungal communities when biogas slurry and chemical fertilizer are applied independently, as well as in combination? (2) What effects do the various treatments have on the chemical characteristics of the soil? This research can offer a crucial theoretical foundation for the rational use of chemical fertilizers and biogas slurry. This study provides recommendations to manage the soil to promote soil health and productivity, while simultaneously minimizing environmental impacts.

## 2. Materials and Methods

### 2.1. Description of the Study Field and Experimental Design

The field used for this study was established at the Sichuan Academy of Agricultural Sciences Farm in Hongya, Sichuan Province, China (29°32′ N, 103°15′ E), and the soil in this area was classified as yellow loam (Appendix A). The annual average temperature is 18 °C, with an annual precipitation of 1663 mm. Silage maize was sown on the site in May, followed by an annual ryegrass (*Lolium multiflorum* L.) rotation in October of each year. After the annual ryegrass was mowed, manual tillage was conducted in plots to a depth of approximately 20 cm before sowing the silage maize. The study utilized a randomized complete block designed for four treatments with three replications. There was a total of 12 plots, and each plot in this study was 28.85 m^2^ (5.95 m × 4.85 m). The initial chemical properties of the soil before the field experiment were as follows: pH = 6.01; soil organic carbon (SOC) 15.31 g kg^−1^; total nitrogen (TN) 1.8 g kg^−1^; total phosphorus (TP) 0.5 g kg^−1^; total potassium (TK) 22.5 g kg^−1^. The plots were separated by concrete walls. The experimental plots were established in 2016, and the crops were rotated between annual ryegrass and silage maize each year for the subsequent 5 years.

The area is primarily dominated by livestock production. Farmers sell annual ryegrass and silage maize to ranches that specialize in cow and cattle farming. The biogas slurry produced by the fermentation of cow and cattle manure is transported to the farmland, creating a sustainable cycle. The biogas slurry used in this experiment was gathered from the Modern Animal Husbandry Hongya Ranch, a commercial biogas plant in Hongya, Sichuan Province, China, which produces biogas by fermenting cow and cattle manure. The biogas slurry was delivered to the test site through a unique pipeline from the biogas plant. The following represent the chemical characteristics of the biogas slurry that was applied: pH = 7.7; organic C: 9380 mg L^−1^; total nitrogen (TN):2660 mg L^−1^; available phosphorus (AP): 15.232 mg L^−1^; available potassium (AK): 28.544 mg L^−1^. The chemical fertilizers tested were urea (46.4%), superphosphate (12%), and potassium chloride (60%), and they were applied at a ratio of N:P:K = 15:5:5.

There were four distinct treatments utilized in this study: (C) control, where no chemical fertilizers or biogas slurry were added; (S) 150 t ha^−1^ biogas slurry; (F) 225 kg ha^−1^ chemical fertilizer; (S-F) 49.5 t ha^−1^ biogas slurry + 150 kg ha^−1^ chemical fertilizer. The biogas slurry replaced one-third of the chemical fertilizer. The study was initiated in 2016. Each year, before sowing the annual ryegrass and silage maize, the plots were plowed. Annual ryegrass and silage maize were sown using the drill sowing method. After the annual ryegrass and silage maize crops grew to the stem-elongation stage, quantitative amounts of biogas slurry and fertilizers were sprayed evenly onto the ground of four different treatment plots. All the chemical fertilizers and the biogas slurry were spread simultaneously. During the growth stages of annual ryegrass and silage maize, the plots were manually weeded to control weed growth.

### 2.2. Analyses of the Soil Samples and Biogas Slurry

The continuous field treatment spanned five years, and the soil samples were gathered from a depth of 0–20 cm on 9 December 2021. To ensure that the soil samples were representative, samples from each plot were collected using a diagonal sampling method, and the samples from three points in each sample plot were mixed into one soil sample. They were then mixed evenly after any visible stones, roots, or debris had been removed. The soil was sieved (≤2 mm), and then the 4 treatment samples were formed into 3 replicate samples per treatment, with 12 soil samples in total. Each sample was divided into 3 sections, from which 5 g was taken out and promptly stored at −80 °C to extract the DNA. Then, 100 g of soil was weighed and stored at 4 °C to examine the microbial biomass C (MBC) and microbial biomass N (MBN), and the remaining sample was air-dried to examine the untreated soil’s chemical properties. Finally, the collected biogas slurry was placed in 100 mL microtubes to determine the chemical indices.

An automated TOC analyzer (TOC-VCPH; Shimazu, Kyoto, Japan) using fumigation based on extraction with chloroform (CHCl_3_) and 0.5 M potassium sulfate (K_2_SO_4_) was employed to ascertain the MBC and MBN [38]. The soil pH was measured with a pH meter at a 1:2.5 ratio (soil:water (w:v)). The TN was determined using an automatic Kjeldahl nitrogen analyzer. The SOC contents were determined with a Vario Max CN analyzer (Elementar Analysensysteme GmbH, Hanau, Germany) [39]. The soil AP was extracted with 0.5 M sodium bicarbonate (NaHCO_3_) (pH = 8.5) and then quantified using molybdenum blue. The amount of AK was determined by flame photometry on extracts of 1 M of ammonium acetate (NH_4_CH_3_CO_2_). In this study, the soil chemical analyses were conducted as previously described [40]. The pH was measured using the glass electrode method in the biogas slurry. Standard techniques were used to determine the amounts of TN, organic carbon, AP, and AK [41].

### 2.3. Extraction of the Soil Genomic DNA, Amplification of the 16S rRNA/ITS Genes, Sequencing, and Data Processing

The genomic DNA was extracted from the soil samples using a FastDNA Spin Kit for Soil (MP Biomedicals, Santa Ana, CA, USA) Reagent Kit. The sample genomic DNA (gDNA) was purified using a Zymo Research BIOMICS DNA Microprep Kit (Cat# D4301; Irvine, CA, USA). The integrity of the DNA was assessed by 0.8% agarose gel electrophoresis, and the concentration was determined using a Tecan F200 (PicoGreen dye method) (Tecan Group, Männedorf, Switzerland).

The 16S rRNA V4 region of the soil bacterial samples was amplified using the following specific primers: 515F (5′-GTGYCAGCMGCCGCGGTAA-3′) and 806R (5‘-GGACTACHVGGGTWTCTAAT-3′). The ITS2 region of the soil fungal samples was amplified using the following primer sequences: ITS3 (5′-GATGAAGAACGYAGYRAA-3′) and ITS4 (5′-TCCTCCGCTTATTGATATGC-3′). The PCR assay was conducted using an Applied Biosystems PCR System 9700 (Applied Biosystems, Waltham, MA, USA). The PCR operation was as follows: pre-denaturation at 94 °C for 1 min in 1 cycle, followed by 25–30 cycles of denaturation at 94 °C for 20 s, annealing at 54 °C for 30 s, and elongation for 30 s at 72 °C, with a final cycle of 5 min at 72 °C and 4 °C heat preservation. Three PCR replicates were performed for each sample, and the PCR products in the linear phase were mixed and used to construct the library. The PCR products were detected using 2% agarose gel electrophoresis. A Zymoclean Gel Recovery Kit (D4008; Zymo Research, Irvine, CA, USA) was used to recover the qualified samples. A Qubit 2.0 Fluorometer (Thermo Fisher Scientific, Waltham, MA, USA) was used for quantification, and the samples were mixed as equal moles. An NEBNext Ultra II DNA Library Prep Kit for Illumina (NEB #E7645L; New England Biolabs, Ipswich, MA, USA) was used to construct the library. The samples were sequenced on an Illumina HiSeq sequencing platform (PE250; Illumina, San Diego, CA, USA) by Rhonin Biotechnology Co., Ltd., Chengdu, China.

### 2.4. Statistical Analysis

The data were organized using Microsoft Excel 2019 (Redmond, WA, USA). The soil chemical data were processed and analyzed using SPSS 26.0 (IBM, Inc., Armonk, NY, USA). R software v. 4.0.5 was used to perform a community composition analysis and alpha-diversity and beta-diversity analyses. The significant differences in soil chemical properties and microbial alpha diversity among treatments were ascertained by a one-way analysis of variance (ANOVA). QIIME2 was used to generate the ASV feature list and feature sequence. A species classification dataset was constructed for the SILVA database using a classifier based on the Naïve Bayes algorithm, and the dataset was used for the species annotation of ASV feature sequences [42]. A differential species analysis was performed using the LEfSe tool [43]. Redundancy analysis (RDA) was performed using the ‘vegan’ package in R 4.3.0 to investigate the relationship between the soil microbial community composition and environmental factors [44]. Principal coordinates analysis (PcoA) analysis based on the Bray–Curtis distance was used to determine beta diversity [45]. Spearman’s rank correlation analyses were performed using the ‘pheatmap’ package in the R software to analyze the correlation between the relative abundance of bacteria and fungi at the genus level and soil chemical properties [46]. Statistical significance between the treatments was defined as *p* < 0.05. Bioinformatic analysis was performed using Omicsmart, a real-time interactive online platform for data analysis (http://www.omicsmart.com, accessed on 14 March 2023).

## 3. Results

### 3.1. Soil Chemical Properties

For 5 years in a row, when the biogas slurry was applied, most of the chemical characteristics of the soil changed significantly when compared to the control (Table 1). For example, the SOC, TN, AP, and AK values increased significantly (*p* < 0.05) by 45.93%, 39.52%, 174.73%, and 161.54%, respectively. However, chemical fertilizers and a combination of biogas slurry and chemical fertilizer treatments did not cause significant changes compared with the control. In addition, compared with the control, the biogas treatment significantly increased soil pH (*p* < 0.05), while the chemical fertilizer treatment significantly decreased soil pH. Compared with the control group, the biogas slurry treatment increased the content of MBC in the soil, while treatment with chemical fertilizers reduced its content in the soil. However, this difference was not significant. The application of chemical fertilizers can significantly reduce the MBN, and a combined biogas slurry and chemical fertilizer treatment can significantly increase this parameter.

### 3.2. Diversity and Composition of the Soil Microbial Communities

The α-diversity indices, including the observed species, Chao, ACE, Shannon, and coverage diversity estimator (Table 2), were used to represent the diversity and richness of the soil bacterial and fungal communities in the three treatments. First, the coverage diversity indicators for the bacterial and fungal communities in each treatment exceeded 97% and 99%, respectively, which indicated that the 16S rRNA and ITS sequences that had been identified could represent the majority of bacteria and fungi present in the samples. The observed species, Chao1, and ACE diversity estimators were significantly higher in the chemical fertilizer treatment compared to the other treatments and the control group. The combined biogas slurry and chemical fertilizer treatment did not differ significantly compared to the control group. There were no significant differences in the observed species, Chao1, Shannon, and ACE indices of the fungal communities among all the treatments.

The Venn diagram analysis Indicated that there were 224 overlapping bacterial operational taxonomic units (OTUs) among all the treatments (Appendix A). The chemical fertilizer treatment had the highest number of unique bacterial OTUs with 303. A total of 42 overlapping fungal OTUs were detected among all the treatments. The chemical fertilizer and combined biogas slurry and chemical fertilizer treatment samples had a higher number of unique fungal OTUs, with 64 and 44, respectively, and they shared a substantial number of fungal OTUs, with a total of 85.

### 3.3. Soil Microbial Community Composition

The bacterial and fungal communities at the phylum and genus levels in the soils subjected to the different treatments are presented in Figure 1 and Figure 2. The analysis focused on the top 14 phyla and genera of bacteria and fungi. In the bacterial community, the most abundant phyla across all the treatments were *Acidobacteria*, *Proteobacteria*, *Bacteroidetes*, *Chloroflexi*, and *Planctomycetes*. The dominant phyla in the control treatment were *Acidobacteria* and *Chloroflexi*, with relative abundances of 29.55% and 13.6%, respectively, which were higher than those in the other treatment groups. In contrast, *Proteobacteria* and *Bacteroidetes* had relative abundances of 19.91% and 8.21%, respectively, which were lower than those in the other treatment groups. In addition, the chemical fertilizer treatment displayed the highest relative abundance of *Proteobacteria* (23.76%) (Figure 1a and Figure 2a). At the genus level, the microbial compositions in the four soil samples were diverse. The top five genera across all the treatments were *Sphingomonas*, *RB41*, *Haliangium*, *Nitrospira*, and *aDurb.Bin063-1*. The abundances of *Haliangium*, *Nitrospira, Gemmatimonas*, and *Lactobacillus* were lower than those in the control and in the biogas slurry treatment, and the abundances of *Flavobacterium* and *Dongia* were higher than those in the control (Figure 1b and Figure 2b).

*Ascomycota*, *Basidiomycota*, and *Mortierellomycota* were the major phyla in the fungal community (Figure 1c). Regarding fungal composition at the phylum level, in the chemical fertilizer treatment, *Ascomycota* had the lowest relative abundance among the four soil systems, while *Basidiomycota* had the highest. In contrast, with the combined biogas slurry and chemical fertilizer treatment, *Ascomycota* had the highest relative abundance, and *Basidiomycota* had the lowest relative abundance (Figure 2c). At the genus level, *Ascobolus*, *Beauveria*, *Plectosphaerella*, and *Emericellopsis* were the top four genera (Figure 1d), but their proportions varied in the soils that were treated in different ways. *Ascobolus* had the highest relative abundance in the biogas slurry treatment group among all the treatments, while *Beauveria* had the lowest relative abundance. *Beauveria, Mortierella*, *Mycena*, and *Talaromyces* had lower abundances in the biogas slurry treatment than in the other treatments and the control group. *Beauveria* was the most abundant fungal genus in the control group and surpassed all the other treatment groups. Conversely, *Emericellopsis* in the chemical fertilizer treatment group had a higher relative abundance compared with the other treatments and the control (Figure 2d).

### 3.4. The Structure and Function of the Soil Microbial Community

The results of the PCoA test for the bacterial and fungal communities are presented in Figure 3. The principal coordinates of the bacterial community (PCoA) (Figure 3a) explained a total of 41.12% of the total variance of the sample composition. The first axis explained 24.16% and the second 16.96%. The results of PCoA of the fungal communities (Figure 3b) showed that the principal coordinates PCo1 and PCo2 explained 20.45% and 14.69% of the total variance for the samples, respectively. The results demonstrated a distinct separation among the bacterial and fungal communities from the control, biogas slurry, chemical fertilizer, and combined biogas slurry and chemical fertilizer treatments. This indicates that the composition of the soil bacterial and fungal communities differed between the treatments.

Redundancy analysis (RDA) was used to analyze the relationship between the soil microbial community and environmental factors in the different treatments. The RDA indicated that the first and second axes collectively explained 36.82% and 31.46% of the total variance for bacteria and fungi (Figure 3c,d), respectively. The positive and negative correlations between the soil microbial community and various environmental factors in each treatment were studied in more detail. The results indicated that the bacterial and fungal community structure was significantly correlated with the pH, TN, SOC, and AP values, while there was a weak correlation between the other soil properties and the microbial community structure. Furthermore, the RDA revealed that the pH, TN, SOC, and AP values represented important environmental factors that influenced the composition of the soil bacterial and fungal community.

### 3.5. Relationships between the Soil Microorganisms and Soil Parameters

In this study, Spearman correlation heatmap analysis was utilized to study the relationship between the soil chemical properties and soil microbial communities. Within the bacterial community, the AP, TN, and SOC positively correlated with *Flavobacterium* and negatively correlated with *Haliangium*. The pH negatively correlated with *Haliangium*, *Nitrospira*, and *Gemmatimonas*. The TN positively correlated with *Dongia,* and the MBN positively correlated with *Lactobacillus* (Figure 4a). Among the fungal community, the pH and AP negatively correlated with *Mortierella* and *Mycena*. The TN and SOC negatively correlated with *Beauveria*, while the TN and SOC positively correlated with *Neosetophoma*. In addition, the AK negatively correlated with *Talaromyces* (Figure 4b). Therefore, the pH, TN, SOC, AP, and AK are important environmental factors that affect the structure of soil bacterial and fungal communities in the annual ryegrass-silage maize rotation.

## 4. Discussion

### 4.1. Effects of Biogas Slurry on the Soil’s Chemical Characteristics

Soil fertility is a crucial indicator that reflects the productivity of plants, and fertilization is a key measure to enhance soil fertility [47]. In this study, after five years of implementing different strategies of fertilization, significant changes were observed in the soil’s chemical properties. The soil SOC, TN, AP, and AK levels increased significantly as a result of the application of biogas slurry (Table 1). The application of biogas slurry resulted in an upward trend in the soil’s microbial MBC and MBN, with a higher content of microbial biomass (MBC and MBN) compared with the control. Chemical fertilizers reduced the soil microbial MBC and MBN. This is consistent with the findings reported from previous research [16,48,49]. MBC in the biogas slurry treatment samples combined with chemical fertilizers was similar to that in the control. This is primarily attributed to the fact that biogas slurry comprises organic wastewater that is produced after anaerobic digestion. Therefore, it contains substantial amounts of organic matter and nutrients, which results in its wide use as a fertilizer [50]. Biogas slurry provides a rich source of C, which favors microbial growth and reproduction, thereby enhancing the microbial biomass in soil [51]. Some studies have found that the application of biogas slurry stimulated soil bacteria and fungi to participate in the SOC cycle; the recalcitrant organic carbon (ROC) in the biogas is decomposed by the soil in the surface layer of the soil, thus contributing to an increase in the soil organic carbon (SOC) stock [52]. Therefore, the application of biogas slurry in this study led to a substantial increase in the content of SOC. Some studies have found that an increase in easily decomposable organic C likely contributed to the enhanced microbial biomass values [53]. Therefore, the increase in soil SOC content under biogas slurry treatment may be one of the reasons for the elevation of MBC and MBN in this study.

Soil acidification has become a major problem in the agricultural system of China, and the amelioration of soil acidification is currently an urgent and pressing challenge [54]. In this study, the experimental site had acidic soil. Compared to the control, the application of chemical fertilizer considerably reduced the soil pH. The acidification of the soil was identified as primarily resulting from the deposition of N, owing to the intensive application of N fertilizer. In particular, the addition of ammonium nitrate (NH_4_NO_3_) and urea made the soil more acidic [55]. Nitrogen fertilizers had been applied to the area for five consecutive years, and the plant roots absorbed ammonium (NH_4_^+^) ions and released H^+^ ions into the soil solution, which resulted in soil acidification [56]. In addition, higher local precipitation may promote acidification. This study suggests that the continued application of fertilizers in this area will further exacerbate soil acidification, to the detriment of soil health. The significant increase in the pH of soil after the application of biogas slurry indicates that biogas slurry can mitigate soil acidification. The primary reason that the application of biogas slurry can alleviate soil acidification is owing to the alkaline nature of the biogas slurry, meaning that the alkaline substances in the biogas slurry can neutralize the soil acidity to some extent. The results of this indicate that the long-term application of biogas slurry can ameliorate the acidic soils to some degree. Therefore, we suggest that acidic soils caused by the long-term use of chemical fertilizers can be improved by the application of biogas slurry, which not only diminishes the reliance on chemical fertilizers and lowers the cost of chemical fertilizers but also neutralizes acidic soil and ensures the health of forage grass soil.

### 4.2. Effects of the Biogas Slurry on the Soil Microbial Community

The diversity and abundance of the microbial communities are reflected by the alpha-diversity indices [57]. In this study, the sequencing data showed that the coverage of bacteria and fungi in all the treatments exceeded 97% and 99%, respectively, and, thus, effectively represented the actual situation of bacterial and fungal communities. A Venn diagram indicated that there were more unique species of soil bacteria and fungi following the application of chemical fertilizer. In this study, the fertilizer treatments showed higher relative abundances of *Nitrospira*, *Gemmatimonas*, *Mortierella*, *Mycena*, and *Torula* at the level of bacterial and fungal genera (Figure 2c,d), which were negatively correlated with pH, compared to the other treatments and the control (Figure 4a,b). The decrease in pH led to an increase in the number of these specific species. Some studies have found that biogas slurry can increase the alpha-diversity indices of bacteria and fungi [58]. There was no significant change in the diversity of soil bacterial and fungal communities in the biogas slurry treatment compared with the control. As shown mainly in the Venn plot (Appendix A), the number of OUTs specific to the biomass slurry bacteria and fungi was similar when compared to the control. In this study, the impact of chemical fertilizer on the alpha-diversity indices of the soil’s bacterial communities was more pronounced than that on the fungal communities. The study found that the application of biogas slurry significantly increased the AP in the soil. Research has also found that long-term high-P fertilizer input increased SOC, MBC, and AP contents, but decreased the total bacterial diversity [59]. This suggests that there may be an indirect negative relationship between the biogas slurry and soil bacterial communities [60]. Part of the reason may be that the bacterial communities are more competitive under nutrient-enriched conditions, which enables them to potentially gain some advantages in their interactions with the fungal communities.

At the bacterial level, these results indicated that *Proteobacteria*, *Acidobacteria*, *Bacteroidetes*, and *Chloroflexi* were the dominant bacterial phyla across all the treatments. Proteobacteria can enhance nutrient cycling in the soil and are highly effective at degrading nutrients [61]. *Acidobacteria* can improve the performance and productivity of crops owing to its role in the N cycle [62]. These bacterial phyla are crucial for the decomposition of materials, such as cellulose and lignin, in the root residues of annual ryegrass and silage maize, the breakdown of soil organic matter, and the utilization of N. Consequently, they contribute significantly to enhancing soil fertility. Research has shown that *Flavobacteria* can degrade large organic molecules, such as proteins and lipids, and possess certain abilities regarding nitrification and potential denitrification [63], which are crucial for natural N cycling [64]. In the soils treated with biogas slurry, the relative abundance of *Flavobacteria* was elevated compared to the control (Figure 4), and there was a strong positive correlation between *Flavobacteria* and the SOC. The results indicate that the application of biogas slurry increases the abundance of the bacteria involved in N cycling in the annual ryegrass soil. The source of soil C is a crucial factor that influences the composition of bacterial communities [65].

The structure of soil microbial communities is influenced by soil parameters [66]. The structure of soil microbial communities is largely determined by the pH of the soil, which can also impact the composition of the microbial communities [67]. In addition, the abundance of bacteria (*Haliangium*, *Nitrospira*, and *Gemmatimonas*) and fungi (*Mortierella* and *Mycena*) after chemical fertilization were higher than with other treatments and in the control. Thus, the decrease in pH owing to fertilization indirectly led to changes in the structure of some bacterial and fungal communities. Previous research has shown that the application of cattle manure alters the chemical properties of the soil, thereby regulating the structure of soil microbial communities [15]. The RDA results showed that the soil’s chemical properties significantly correlated with the bacterial and fungal communities after different treatments. The structure of bacterial communities significantly correlated with certain soil parameters, such as pH, SOC, TN, AP, and MBN, and also played a role in the formation of the fungal community structure. Additionally, a Spearman correlation study revealed a significant correlation between these soil parameters and the individual microbial communities of bacteria and fungi. In conclusion, we hypothesized that variations in the composition of the soil microbial community could be connected to the accumulation of soil nutrients.

While numerous studies have demonstrated that the introduction of exogenous microorganisms can alter the diversity and structure of microbial communities [68], they generally focused on how a single microbial inoculant affects the microbial communities in soil [69]. For this study, the overall results indicated that the relative abundances of some bacterial and fungal communities in the soil were altered by the application of biogas slurry, but the higher relative abundance of microorganisms introduced via the biogas slurry, such as *Firmicutes*, *Fastidiosipila*, *Ascomycota*, and *Plectosphaerella*, did not seem to affect the soil microorganisms (Appendix A). However, owing to the complex microbial composition of the biogas slurry, there is no evidence to support the concept that the introduction of microorganisms from the biogas slurry changed the soil’s microbial diversity and community structure. We hypothesize that the introduction of microorganisms from biogas slurry is not the primary factor that drives changes in soil microbial diversity and communities. Perhaps microorganisms released from the biogas slurry into the soil environment are weaker competitors compared to the indigenous soil microbiota, or perhaps some of the unique microorganisms in the biogas slurry find it difficult to adapt to the soil environment, which renders the impact of introduced microorganisms from the biogas slurry negligible. Similarly, one study found that the bacterial community in the soil was not significantly impacted by microorganisms derived from the composted organic matter [70]. However, specific conclusions require further research and validation.

### 4.3. Sustainability of Soil Health with the Application of Biogas Slurry Compared to That of Chemical Fertilizer

Soil health is the foundation of soil sustainability, and a healthy soil environment is more likely to result in the utilization of sustainable soil. In this study, there were no appreciable changes in the soil SOC, TN, AP, and AK values following the application of chemical fertilizer compared to the control (C). The reason for this could be that the fertilizer provided just enough nutrients to meet the plant’s growth requirements and did not cause a build-up of the nutrients. Another reason could be the loss and leaching of nutrients due to rainfall, which results in the transport of nutrients to deeper soil layers, thereby reducing the levels of nutrients in the surface soil. There were no significant changes in the contents of soil nutrients compared to the control group (C), following the combined application of biogas slurry and chemical fertilizer. This result contrasts with those of others [60]. We hypothesized that one of the reasons for this difference could be the lower amount of chemical fertilizer applied owing to its replacement with biogas slurry, and further research is merited to determine the specific reasons. In future research, we will identify the optimal ratio for combining the biogas slurry and chemical fertilizer to reduce the usage of chemical fertilizer and input costs, while maintaining the levels of soil nutrients at reasonable levels. In addition to this, a significant amount of heavy metals are present in the biogas slurry, which increases the possibility that heavy metals will accumulate in the soil following repeated applications of biogas slurry [71,72]. These heavy metals can be transferred to the food chain through the forage grass and could pose a potential risk from livestock products. When biogas slurry is applied on a more long-term basis, a risk assessment of heavy metals in the pasture after the application of biogas should be conducted appropriately. In southwest China, where animal husbandry is well developed, livestock produce a large amount of biogas slurry every year, which is consumed locally by returning the biogas slurry to the field and planting annual ryegrass. This approach not only reduces the cost of fertilizer use but also creates a self-sustaining cycle of soil–forage grass–livestock.

## 5. Conclusions

After five years of applying biogas slurry, chemical fertilizer, and a combination of both to a rotation of annual ryegrass-silage maize, this study identified alterations in the chemical characteristics and microbial communities of the soil. Compared with the control group, this study showed that the application of biogas slurry did not significantly increase the diversity of soil bacteria, but effectively regulated the structure of soil bacterial and fungal communities. No differences in fungal community diversity were observed among all the treatments. The application of biogas slurry increased the levels of soil organic carbon (SOC), total nitrogen (TN), available phosphorus (AP), and total potassium (AK) in the soil, thereby enhancing its fertility. It also neutralizes acidic soils and increases the soil pH. The TN, SOC, pH, and AP values are all important environmental factors that influence the composition of soil bacterial and fungal communities. This study offers theoretical reinforcement for the proper and enduring use of biogas slurry, which is vital for lessening the usage of chemical fertilizers, implementing appropriate fertilization procedures, and upholding the ecological balance of agricultural soils.

## Figures and Tables

**Figure 1 microorganisms-12-00716-f001:**
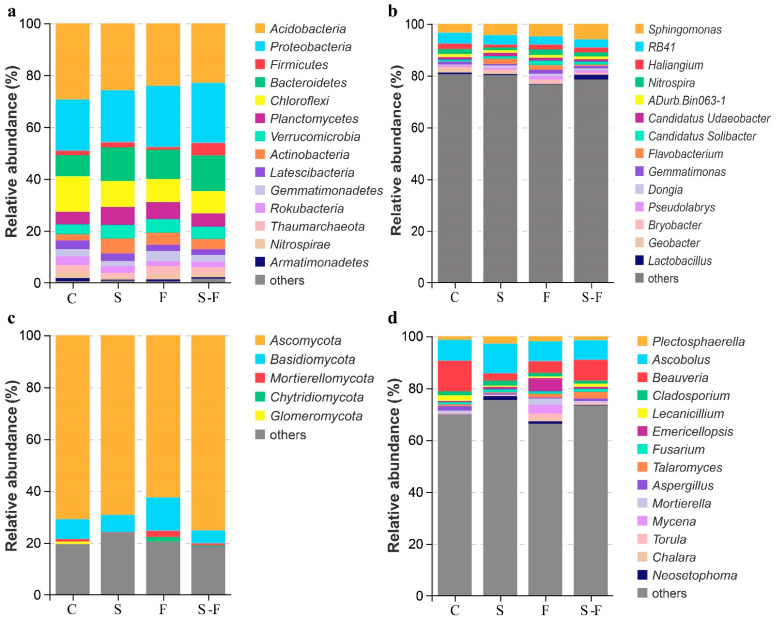
Relative abundance of microbial communities under different treatments. (**a**) Bacteria phylum; (**b**) bacteria genus; (**c**) fungi phylum; (**d**) fungi genus. C, control; S, biogas slurry; F, chemical fertilizers; S-F, biogas slurry combined with chemical fertilizers.

**Figure 2 microorganisms-12-00716-f002:**
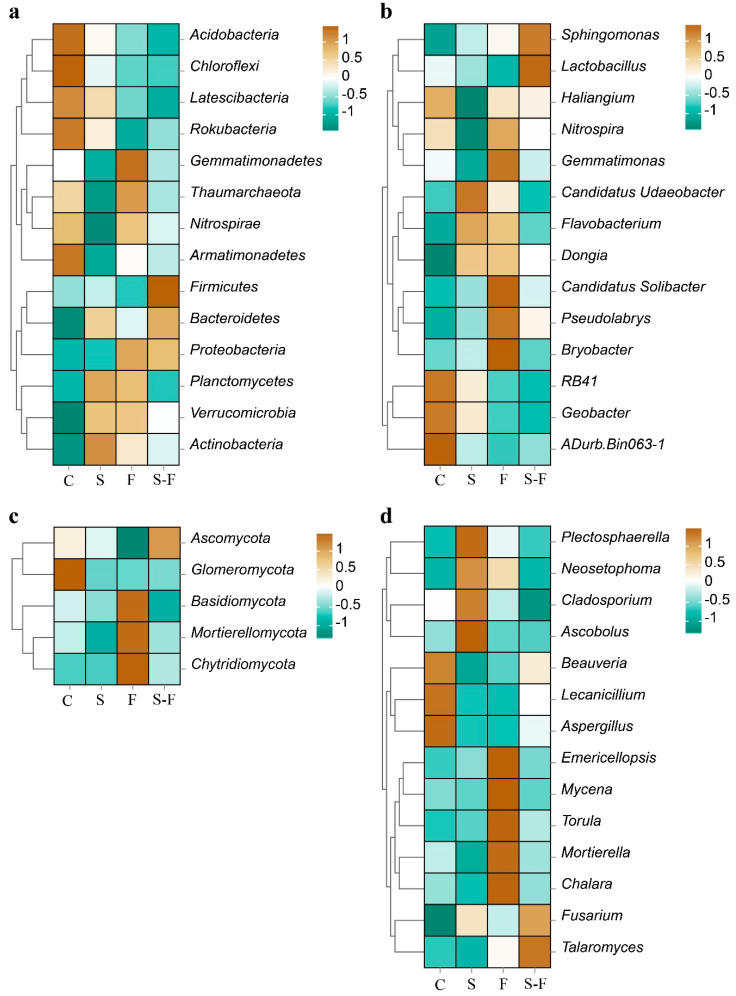
Heatmap of relative abundance of microbial communities. (**a**) Bacteria phylum; (**b**) bacteria genus; (**c**) fungi phylum; (**d**) fungi genus. C, control; S, biogas slurry; F, chemical fertilizers; S-F, biogas slurry combined with chemical fertilizers. The colors in the heatmap represent the abundance shown in the legend; the darker the color, the higher the abundance.

**Figure 3 microorganisms-12-00716-f003:**
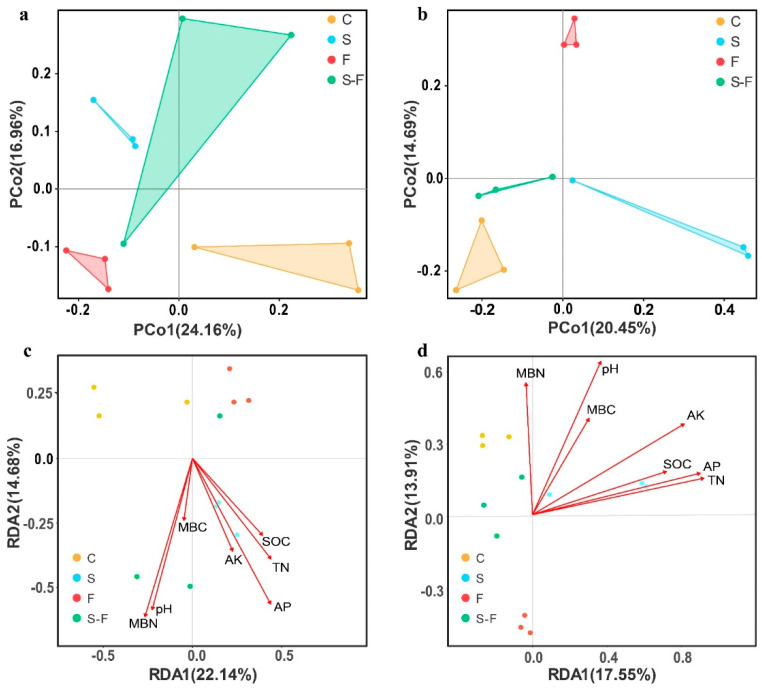
Principal coordinate analysis (PCoA) and redundancy analysis (RDA) results. Principal coordinate analysis (PCoA) results were based on the Bray–Curtis distances of bacterial (**a**) and fungi (**b**) communities for soil undergoing different treatments. Redundancy analysis (RDA) results, showing the correlation between microbial communities ((**c**), bacteria and (**d**), fungi) and environmental factors. C, control; S, biogas slurry; F, chemical fertilizers; S-F, biogas slurry combined with chemical fertilizers.

**Figure 4 microorganisms-12-00716-f004:**
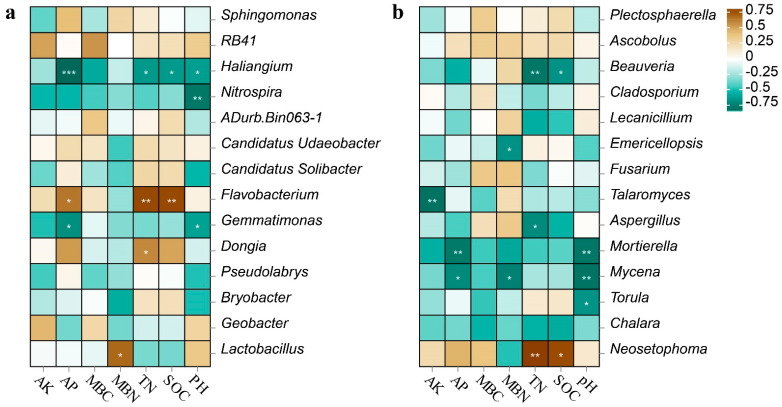
Spearman correlation heatmap between soil bacterial (**a**) and fungal (**b**) communities and soil chemical characteristics at the genus level. Colors in the heatmap represent correlations, as indicated by the legend, with positive correlations in brown and negative correlations in blue. * *p* < 0.05, ** *p* < 0.01, *** *p* < 0.001.

**Table 1 microorganisms-12-00716-t001:** Effects of different treatments on the chemical constituents of the soil.

Attribute	C	S	F	S-F
SOC (g/kg)	14.15 ± 0.21 b	20.65 ± 1.09 a	14.4 ± 0.62 b	13.81 ± 0.75 b
TN (g/kg)	1.67 ± 0.04 b	2.33 ± 0.06 a	1.73 ± 0.06 b	1.68 ± 0.04 b
C/N	8.47 ± 0.10 a	8.91 ± 0.70 a	8.33 ± 0.09 a	8.22 ± 0.28 a
AP (mg/kg)	27.23 ± 4.18 b	74.81 ± 6.19 a	30.87 ± 3.83 b	36.02 ± 1.83 b
AK (g/kg)	0.13 ± 0.02 b	0.34 ± 0.04 a	0.07 ± 0.01 b	0.07 ± 0.004 b
MBC (mg/kg)	196.4 ± 12.13 ab	228.78 ± 12.94 a	171.37 ± 20.29 b	196.13 ± 9.88 ab
MBN (mg/kg)	17.10 ± 3.23 b	20.32 ± 1.45 b	6.65 ± 0.97 c	31.81 ± 1.60 a
pH	5.68 ± 0.02 b	5.83 ± 0.02 a	5.43 ± 0.05 c	5.67 ± 0.02 b

Note: Values are the mean ± SE (n = 3). Different lower-case letters within the same row denote significant (*p* < 0.05) differences between the treatments. C, control; S, biogas slurry; F, chemical fertilizers; S-F, biogas slurry combined with chemical fertilizers.

**Table 2 microorganisms-12-00716-t002:** Community diversity indices of soil microbial.

	Treatment	Observed Species	Chao1	Shannon	ACE	Coverage
Bacteria	C	1204 b	1270.31 b	6.43 a	1282.35 b	0.982
	S	1246 ab	1352.38 ab	6.47 a	1356.49 b	0.977
	F	1357 a	1498.41 a	6.53 a	1519.71 a	0.970
	S-F	1188 b	1287.23 b	6.41 a	1283.27 b	0.979
	*p*-value	*	*	ns	*	
Fungi	C	244 a	262.63 a	3.87 a	261.84 a	0.996
	S	272 a	296.04 a	3.82 a	295.14 a	0.994
	F	270 a	281.05 a	4.28 a	282.57 a	0.996
	S-F	263 a	281.73 a	4.21 a	276.43 a	0.996
	*p*-value	ns	ns	ns	ns	

The mean values of different letters in the same column (n = 3) indicate significant differences among the treatments (*p* < 0.05). C, control; S, biogas slurry; F, chemical fertilizers; S-F, biogas slurry combined with chemical fertilizers. “*” indicates the statistical significance at *p* < 0.05, and “ns” indicates no significant difference.

## Data Availability

The raw data supporting the conclusions of this article have been deposited at the NCBI (https://www.ncbi.nlm.nih.gov, accessed on 18 October 2023) with BioProject: PRJNA1029594 (accessions: SRR26421992, SRR26421989, SRR26421986, SRR26421983, SRR26421980, SRR26421993, SRR26421990, SRR26421987, SRR26421984, SRR26421981, SRR26421988, SRR26421985, SRR26421982, SRR26421979, SRR26421994, and SRR26421991) and BioProject: PRJNA1029597 (accessions: SRR26422091, SRR26422088, SRR26422085, SRR26422082, SRR26422079, SRR26422092, SRR26422089, SRR26422086, SRR26422083, SRR26422080, SRR26422077, SRR26422090, SRR26422087, SRR26422084, SRR26422081, and SRR26422078).

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
