# Peer review of "Effect of Biogas Slurry on the Soil Properties and Microbial Composition in an Annual Ryegrass-Silage Maize Rotation System over a Five-Year Period"

_microorganisms, 2024, doi:10.3390/microorganisms12040716_

Round 1

Reviewer 1 Report

Comments and Suggestions for Authors

This is a novel study that investigates long-term (5 year) effects of addition  of biogas slurry, mineral fertilizer and a combination of both on soil chemistry and microbiology compared to controls without the additions.

On the whole the paper is useful, and fairly well written, but has some serious deficiencies in the abstract and in the discussion in drawing conclusions about the effects of the biogass slurry on the diversity of the  bacterial communities - where the only treatment to show statistically significant differences to the control is the fertilizer only treatment (Table 2). 

The paper claims to report physicochemical properties of the soils, but in fact the paper only reports chemical and microbiological results and all mention of physical and "physio" chemical should be removed.

The paper would benefit from careful editing throughout to provide more essential details regarding methodology, and correct some relatively minor English language and text formatting errors.  The discussion is too long and too repetitive and could be usefully condensed.

 I attach a marked up file showing the areas of greatest concern (red highlights) and other points for revision.

 Overall, the findings are of interest in showing that for the data generated by these microbiological community analysis approaches there is a surprising lack of substantial effect of the slurry on the soil microbiomes even after 5 successive years of treatment. Possibly in these arable-grassland cropping systems the effects of the field management practices and the crop species themselves provide a major control on the microbial communities, and the slurry additions have little effect relative to the other environmental drivers of the communities.

Comments on the Quality of English Language

Author Response

Dear Theodore Tang,

Thanks for your hard work on our manuscript. You and the other two reviewers’ comments are all of great importance to our manuscript. Based on the reviews, we have revised our manuscript point-by-point and explains the revisions we have made as following. All the revisions were highlighted in the revised manuscript.

Reviewer #1

  1. Annotation “Which fertilizer- what rates?”, “Biogas feedstock?” and “Amounts?” line 16 to line 17 page 1.

Response: We have revised “This study aimed to investigate the impacts of applying chemical fertilizer, biogas slurry, and their combination for 5 years on the physicochemical properties and microbial community of farmed soils” as “This study aimed to investigate the impacts on the chemical properties and microbial community of farmland soils which was applied chemical fertilizer (NPK) (225 kg ha-1), biogas slurry (150 t ha-1), and their combination (49.5 t ha-1 biogas slurry + 150 kg ha-1 chemical fertilizer) for five years” line 16 to line 18.

  1. Annotation “data”, “Functional benefits?” line 19 to line 23 page 1.

Response: We have modified the descriptions as “The results indicated that compared to the control group, the long-term application of biogas slurry significantly increased the SOC, TN, AP, and AK by 45.93%, 39.52%, 174.73% and 161.54%, respectively; neutralized acidic soil and increased the soil pH” line 19 to line 21.

  1. Annotation “This statement contradicts the data evidence that shows that diversity was unchanged by slurry compared to the control, and was less than with mineral fertilizer additions so this conclusion is wrong.” line 25 page 1.

Response: Thanks for your review. We have removed this incorrect conclusion. We have rewritten the conclusions as “Chemical fertilizer application significantly increased the diversity of the bacterial community.” line 23 to line 24.

  1. Annotation “References”, “To what extent?” “Data?” line 40 to line 46 page 2.

Response: Modifications have been made line 40 to line 45.

  1. Annotation “from what feedstocks?” line 50 page 2

Response: We have added “Biogas slurry is mainly the high-concentrated organic wastewater produced by anaerobic fermentation of organic matter, such as livestock and poultry manure” line 48 to 49, and details line 119 to 120.

  1. Annotation “Data define”, “Define?” line 57 and line 62 page 2

Response: Revisions have been made as “However, previous research has shown that compared with the application of low-concentration biogas slurry (250 m3 ha-1 yr-1), the application of high-concentration biogas slurry (375 m3 ha-1 yr-1) had the highest alpha diversity of fungal community, but not the bacterial diversity. Furthermore, the application of biogas slurry also significantly influenced the abundance of functional fungi such as Aspergillus, Trichoderm and Penicillium” line 56 to 60.

  1. Annotation “Does not follow. Reference needed.”, “What kinds of livestock? What volumes of manure?” line 66 to line 67 page 2
  2. Annotation “Data, Evidence, References?” line 69 to line 70 page 2

Response: We have revised the description as “Alternatively, China is the 5th largest livestock producer in the world and it is estimated that over 551 Mt of manure (dry weight base) is generated in China per year. Therefore, the effective utilization of biogas slurry need further attention [20,21]. The application of biogas slurry, rather than chemical fertilizer, can solve the issue of excessive manure produced by animal husbandry. Additionally, it is recognized as a key factor in achieving the sustainable biogas slurry-forage grass-livestock production cycle [22-24].” line 66 to line 72.

  1. Annotation “Not clear?” line 74 page 2

Response: We have revised the description as “However, improper application of biogas slurry may cause soil salinization and heavy metals accumulation, posing a potential pollution risk to farmland soil [25]. Therefore, it is necessary to study the effects of biogas slurry application on forage planting soil” line 72 to line 75.

  1. Annotation “such as?” line 78 page 2

Response: We have revised the description “As an essential part of the soil ecosystem, soil microorganisms are involved in the decomposition of animal waste and plants [26], the formation of humus and the transformation and cycling of nutrients [27,28]” line 76 to line 78.

  1. Annotation “Soil health and microbial diversity are typically not related more diverse communities are common in degraded soils- due to less dominance” line 79 to 80 page 2

Response: We have revised sentence as “Soil microorganisms play a crucial role in maintaining soil fertility and soil microbial communities which are the indicators for evaluating soil health.” Line 78 to 80.

  1. Annotation “Data? Evidence? References?” line 84 to 85 page 2

Response: We have revised sentence as “Application of biogas slurry can increase soil pH and NH4+-N, and mitigate soil acidification and reshape the soil microbial community” line 83 to 85.

  1. Annotation “This arguments is weak and the case overstated.” line 86 page 2

Response: Revised.

  1. Annotation “Evidence? References?” “Is there any? Previous studies should be cited- how many years of applications should be stated.” line 88 to 89 page 3

Response: We have made the revisions as “To date, most studies have focused on the short-term effects of biogas slurry on yield and quality in forage monoculture systems” line 85 to line 86.

  1. Annotation “Argument appears self-contradictory/ not properly explained.” “Define soil health” line 90 to 91 page 3

Response: We have made the revisions as “Therefore, long-term field experiments are needed to thoroughly investigate the changes in soil properties and microbial communities in order to unravel the complexity of biogas slurry application to soil.” Line87 to 89.

  1. Annotation “Texture?” line 110 page 3

Response: We have modified the description as “and the soil in this area was classified as yellow loam” line 105 to line106.

  1. Annotation “was the ryegrass mown? was it herbicide treated or ploughed before the maize was sown?” line 112 page 3

Response: We have added details as “After the annual ryegrass was mowed, manual tillage was conducted in plots to a depth of approximately 20 cm before sowing the silage maize.” line 108 to line 110, and “During the growth stages of annual ryegrass and silage maize, the plots are manually weeded to control weed growth.” line 138 to line 140.

  1. Annotation “What kind of livestock? Cattle, pigs sheep or just cattle?” line 122 page 3

Response: We have made the revisions “The area is primarily dominated by livestock production. Farmers sell annual ryegrass and silage maize to ranches that specialize in cow and cattle farming. The biogas slurry produced by the fermentation of cow and cattle manure is transported to the farmland, creating a sustainable cycle.” line 118 to 121.

  1. Annotation “Based on dry or wet weight?” line 128 page 3

Response: Revised

  1. Annotation “Abbreviations are non intuitive- control - C; slurry- S; fertilizer F” line 132 page 3

Response: We have made the revisions throughout the manuscript.

  1. Annotation “Why are rates per 667 m-2 standardize to ha-1 values to compare with other publications.” line 133 to 134 page 3

Response: We have made the modification line 130 to line 132.

  1. Annotation “How were the plots cultivated? Was this direct driling?” “Were herbicides used?” line 137 to 138 page 3

Response: We have made additions “Each year, before sowing annual ryegrass and silage maize, the plots were ploughed. Annual ryegrass and silage maize were sown by drill sowing method.” line 133 to line 135.

  1. Annotation “How many samples per plot?” line 142 page 4

Response: We have revised “To ensure that the soil samples were representative, samples of each plot were collected using diagonal sampling method, and the samples from three points in each sample plot were mix into one soil sample” and “Three replicate samples per treatment and twelve soil samples in total.” line 147 to line 148.

  1. Annotation “Method details and reference needed.” line 154 page 4

Response: Revised in line 157 to line 158

  1. Annotation “In contrast” line 259 page 4

Response: Modifications have been made line 261.

  1. Annotation “Same data as Figure 1a? Not sure we need both.” page 8

Response: Figure 1 primarily represents the relative abundance of phylum or genus for the top 14 microorganisms across all treatments, but it does not visualize how high or low the relative abundance of each phylum or genus is, especially at the genus level. Therefore, we draw Figure 2 to show relative abundance of species in each treatment.

  1. Annotation “What are the units of the numerical scale?” line 290 page 8

Response: The heatmap displays the normalised relative abundance values of the microbial community under different treatments, therefore, units have not been labelled.

  1. Annotation “The data do not support this interpretation- the biogas slurry is no more influential than mineral fertilizer- and less so with respect to effects on bacterial diversity.” line 312 to 313 page 8

Response: Revised

  1. Annotation “No evidence of this from a diversity perspective.” line 351 page 11

Response: Revised

  1. Annotation “Evidence or speculation?” line 352 page 11

Response: We have added relevant literature.

  1. Annotation “The data presented contradicts this statement” line 388 to 389 page 11

Response: Revised

  1. Annotation “Table 2- slurry shows no increase in bacterial alpha diversity relative to the control-” line 395 page 12

Response: We have revised it as “The study found that the application of biogas slurry significantly increased the AP in the soil. Research has also found that long-term high-P fertilizer input increased SOC, MBC and AP, but decreased the total bacterial diversity” line 400 to 403.

  1. Annotation “Repetitive- this point has been made already” line 416 to 417 page 12

Response: Revised

  1. Annotation “Not really- most of the variation is not explained by the measured soil variables.” line 419 to 420 page 13

Response: We have revised it as “The structure of soil microbial communities is influenced by soil parameters” line 424.

  1. Annotation “Repetitive” line 422 to 423 page 12

Response: Revised.

  1. Annotation “Repetitive” line 424 to 426 page 12

Response: Revised.

  1. Annotation “Repetitive” line 463 to 466 page 12

Response: Revised.

  1. Annotation “How is "sucess" evaluated- there is no evidence of any benefits?” line 499 to 500 page 12

Response: We have modified it as “Compared with the control group, this study showed that the application of biogas slurry did not significantly increase the diversity of soil bacteria, but effectively regulated the structure of soil bacterial and fungal communities” line 490 to 493.

  1. Annotation “Italic” “Italics for Latin names?” “italics for genus names” “Superscript +” line 463 to 466 page 12

Response: We have made corrections in the manuscript according to the annotation.

Thank you again for consideration of our revised manuscripts to be published on Microorganisms.

Sincerely Yours,

Xinquan Zhang

E-mail: zhangxq@sicau.edu.cn

College of Grassland Science and Technology, Sichuan Agricultural University

Huiming Road 211, Wenjiang, Chengdu 611130, Sichuan, China,

Tel: +86 13981616290

Reviewer 2 Report

Comments and Suggestions for Authors

This study analyzed the effect of the long-term application of biogas slurry and chemical fertilizer and their combination on the physicochemical characteristics of the soil and the composition, structure, and diversity of bacterial and fungal communities. The authors found differences in the structure of the bacterial and fungal communities between the treatments but no differences in the diversity metrics. On the other hand, trends were found in soil parameters such as TN, AP, AK, and SOC in the slurry treatment compared to controls and the combination with chemical fertilization.

The MS is well organized, and the experimental design is appropriate. Although the number of replicates was only 3, the authors identified patterns that allowed them to answer the main scientific questions of the study. However, the results do not support some interpretations and conclusions. Regarding the methods, relevant information was not included, and other analyses do not belong to this study (e.g., PCA, NMDS).

Therefore, the MS must improve in the above aspects for publication.

Specific comments L22. Does improving diversity refer to the slurry treatment? The highest diversity was in the treatment with chemical fertilization, although there were no significant differences when compared to the other treatments.

L90-92. Unclear meaning for 'instability of soil microorganisms'. Please clarify.

L54-56. The introduction may benefit from the addition of recent reviews on this topic, so consider the following:

Wang, Z., Sanusi, I. A., Wang, J., Ye, X., Kana, E. B. G., Olaniran, A. O., & Shao, H. (2023). Developments and Prospects of Farmland Application of Biogas Slurry in China—A Review. Microorganisms, 11(11), 2675.

L69-L74. Use consistently 'biogas slurry'.

L97. Confusing statement 'on the composition and physicochemical properties of the soil microbial community'? Rephrase.

L132-137. Doses for slurry and fertilization must be referred to as per hectare.

L141-143. Indicate the number of samples used for soil/microbial analyses.

L148. If 'EP' stands for Eppendorf, replace it with 'microtubes'.

L190-192. Add the basic methods for bioinformatics, trimming, database for taxonomic assignation (OTUS, ASVs), etc.

L193-206. This paragraph contains several errors. 1) Indicate the purpose of the main statistical analyses, and 2) delete all those analyses not performed in this study, for instance, PCA and NMDS. Moreover, this study did not include functional analysis by PICRUST and FAPROTAX. Correct accordingly.

L215. According to what type of statistical analysis? Please revise this carefully and make the necessary corrections.

L307-312. Improve the interpretation of the RDA by adding values supporting variables' correlation to RDA1/RDA2.

L343. It is not clear from which analysis this conclusion is drawn. Looking at Figure 3 (RDA) and Table 1, the variables determining differences among treatments (and therefore an effect on the microbial communities) are TN, AP, SOC, and pH.

L343. At least MBN does not differ significantly from the control. MBC is similar to the control and combined slurry and fertilization treatment. Please revise and interpret correctly.

L384-387. Some of these correlations are not significant; revise.

L388-389. Wrong interpretation. There were no statistical differences in bacterial or fungal diversity.

L450. Please revise the interpretation. First, Fig. S3 requires a caption. One may guess that ZY stands for 'slurry'; however, the higher relative abundance belongs to 'Firmicutes,' not 'Acidobacteria.' Please clarify.

Conclusions: Modify according to corrections in results and discussion.

All figures and tables require an indication of the type of statistical analysis used to obtain such results. For instance, in Fig. 2, what analysis was used to build this heatmap?

Comments on the Quality of English Language

Minor issues detected

Author Response

Dear Theodore Tang, Thanks for your hard work on our manuscript. You and the other two reviewers’ comments are all of great importance to our manuscript. Based on the reviews, we have revised our manuscript point-by-point and explains the revisions we have made as following. All the revisions were highlighted in the revised manuscript. Reviewer #2: This study analyzed the effect of the long-term application of biogas slurry and chemical fertilizer and their combination on the physicochemical characteristics of the soil and the composition, structure, and diversity of bacterial and fungal communities. The authors found differences in the structure of the bacterial and fungal communities between the treatments but no differences in the diversity metrics. On the other hand, trends were found in soil parameters such as TN, AP, AK, and SOC in the slurry treatment compared to controls and the combination with chemical fertilization. The MS is well organized, and the experimental design is appropriate. Although the number of replicates was only 3, the authors identified patterns that allowed them to answer the main scientific questions of the study. However, the results do not support some interpretations and conclusions. Regarding the methods, relevant information was not included, and other analyses do not belong to this study (e.g., PCA, NMDS). Therefore, the MS must improve in the above aspects for publication. 1. Specific comments L22. Does improving diversity refer to the slurry treatment? The highest diversity was in the treatment with chemical fertilization, although there were no significant differences when compared to the other treatments. Response: Thanks for your review. We have revised the describe as “Chemical fertilizer application significantly increased the diversity of the bacterial community” line 23 to line 24. 2. L90-92. Unclear meaning for 'instability of soil microorganisms'. Please clarify. Response: We have modified the sentence as “Therefore, long-term field experiments are needed to thoroughly investigate the changes in soil properties and microbial communities in order to unravel the complexity of biogas slurry application to soil” line 87 to 89. 3. L54-56. The introduction may benefit from the addition of recent reviews on this topic, so consider the following: Wang, Z., Sanusi, I. A., Wang, J., Ye, X., Kana, E. B. G., Olaniran, A. O., & Shao, H. (2023). Developments and Prospects of Farmland Application of Biogas Slurry in China—A Review. Microorganisms, 11(11), 2675. Response: Thanks for the recommendation, this review is beneficial to our introduction part and we have referenced this review in our manuscript line 72 to line 74. 4. L69-L74. Use consistently 'biogas slurry'. Response: Revised. 5. L97. Confusing statement 'on the composition and physicochemical properties of the soil microbial community'? Rephrase. Response: We have revised the describe as “on the soil chemical properties and soil microbial community composition” line 93 to line 94. 6. L132-137. Doses for slurry and fertilization must be referred to as per hectare. Response: We have modified the description line 130 to 132. 7. L141-143. Indicate the number of samples used for soil/microbial analyses. Response: The number of samples used for soil/microbial analysis has been indicated line 143 to 145. 8. L148. If 'EP' stands for Eppendorf, replace it with 'microtubes'. Response: We have replaced 'EP tubes' with 'microtubes' line 152. 9. L190-192. Add the basic methods for bioinformatics, trimming, database for taxonomic assignation (OTUS, ASVs), etc. Response: We have added basic methods for bioinformatics, trimming, database for taxonomic assignation line 195 to 200. 10. L193-206. This paragraph contains several errors. 1) Indicate the purpose of the main statistical analyses, and 2) delete all those analyses not performed in this study, for instance, PCA and NMDS. Moreover, this study did not include functional analysis by PICRUST and FAPROTAX. Correct accordingly. Response: We have indicated the purpose of the main statistical analysis and with modifications line 200 to 207. 11. L215. According to what type of statistical analysis? Please revise this carefully and make the necessary corrections. Response: We performed a one-way ANOVA on soil chemical properties using SPSS software. We have revised the description line 207 to 209. 12. L307-312. Improve the interpretation of the RDA by adding values supporting variables' correlation to RDA1/RDA2. Response: We have modified line 308 to 313 13. L343. It is not clear from which analysis this conclusion is drawn. Looking at Figure 3 (RDA) and Table 1, the variables determining differences among treatments (and therefore an effect on the microbial communities) are TN, AP, SOC, and pH. 14. L343. At least MBN does not differ significantly from the control. MBC is similar to the control and combined slurry and fertilization treatment. Please revise and interpret correctly. Response: The soil TN, AP, and AK increased significantly as a result of the application of biogas slurry. We have made revisions line 344 to line 346. 15. L384-387. Some of these correlations are not significant; revise. Response: We have made revisions line 389 to line 391. 16. L388-389. Wrong interpretation. There were no statistical differences in bacterial or fungal diversity. Response: We have made revisions line 395 to line 396. 17. L450. Please revise the interpretation. First, Fig. S3 requires a caption. One may guess that ZY stands for 'slurry'; however, the higher relative abundance belongs to 'Firmicutes,' not 'Acidobacteria.' Please clarify. Response: We've made modifications based on your review. We have modified 'Acidobacteria' to 'Firmicutes' in line 446, and we also modified the legend of Figure S3. 18. Conclusions: Modify according to corrections in results and discussion. Response: We have revised the results and the discussion line 490 to line 193. 19. All figures and tables require an indication of the type of statistical analysis used to obtain such results. For instance, in Fig. 2, what analysis was used to build this heatmap? Response: We have added the detail in “Statistical analysis” part line 204 to line 209. Thank you again for consideration of our revised manuscripts to be published on Microorganisms. Sincerely Yours, Xinquan Zhang E-mail: zhangxq@sicau.edu.cn College of Grassland Science and Technology, Sichuan Agricultural University Huiming Road 211, Wenjiang, Chengdu 611130, Sichuan, China, Tel: +86 13981616290
